# Differential In Vitro Effects of SGLT2 Inhibitors on Mitochondrial Oxidative Phosphorylation, Glucose Uptake and Cell Metabolism

**DOI:** 10.3390/ijms23147966

**Published:** 2022-07-19

**Authors:** Elmar Zügner, Hsiu-Chiung Yang, Petra Kotzbeck, Beate Boulgaropoulos, Harald Sourij, Sepideh Hagvall, Charles S. Elmore, Russell Esterline, Sven Moosmang, Jan Oscarsson, Thomas R. Pieber, Xiao-Rong Peng, Christoph Magnes

**Affiliations:** 1Institute for Biomedicine and Health Sciences (HEALTH), Joanneum Research Forschungsgesellschaft m.b.H, Neue Stiftingtalstrasse 2, 8010 Graz, Austria; elmar.zuegner@joanneum.at (E.Z.); beate.boulgaropoulos@joanneum.at (B.B.); thomas.pieber@joanneum.at (T.R.P.); 2Research and Early Development, Cardiovascular, Renal and Metabolism, BioPharmaceuticals R&D, AstraZeneca, 431 83 Gothenburg, Sweden; ginnie.yang@bayer.com (H.-C.Y.); sepideh.hagvall@astrazeneca.com (S.H.); sven.moosmang@astrazeneca.com (S.M.); 3Division of Endocrinology and Diabetology, Department of Internal Medicine, Medical University of Graz, Auenbruggerplatz 15, 8036 Graz, Austria; petra.kotzbeck@joanneum.at (P.K.); ha.sourij@medunigraz.at (H.S.); 4Cooperative Centre for Regenerative Medicine (COREMED), Joanneum Research Forschungsgesellschaft m.b.H, Neue Stiftingtalstrasse 2, 8010 Graz, Austria; 5Research Unit for Tissue Regeneration, Repair and Reconstruction, Division of Plastic, Aesthetic and Reconstructive Surgery, Department of Surgery, Medical University of Graz, 8036 Graz, Austria; 6Pharmaceutical Sciences, R&D, AstraZeneca, 415 25 Göteborg, Sweden; chad.elmore@astrazeneca.com; 7Late-Stage Development, Cardiovascular, Renal and Metabolism, BioPharmaceuticals R&D, AstraZeneca, Gaithersburg, MD 20878, USA; russell.esterline@astrazeneca.com (R.E.); jan.oscarsson@astrazeneca.com (J.O.)

**Keywords:** SGLT2 inhibitor, dapagliflozin, empagliflozin, canagliflozin, ertugliflozin, metabolomics, mitochondrial oxidative phosphorylation, glucose uptake, cellular energy level, cell metabolism

## Abstract

(1) The cardio-reno-metabolic benefits of the SGLT2 inhibitors canagliflozin (cana), dapagliflozin (dapa), ertugliflozin (ertu), and empagliflozin (empa) have been demonstrated, but it remains unclear whether they exert different off-target effects influencing clinical profiles. (2) We aimed to investigate the effects of SGLT2 inhibitors on mitochondrial function, cellular glucose-uptake (GU), and metabolic pathways in human-umbilical-vein endothelial cells (HUVECs). (3) At 100 µM (supra-pharmacological concentration), cana decreased ECAR by 45% and inhibited GU (IC5o: 14 µM). At 100 µM and 10 µM (pharmacological concentration), cana increased the ADP/ATP ratio, whereas dapa and ertu (3, 10 µM, about 10× the pharmacological concentration) showed no effect. Cana (100 µM) decreased the oxygen consumption rate (OCR) by 60%, while dapa decreased it by 7%, and ertu and empa (all 100 µM) had no significant effect. Cana (100 µM) inhibited GLUT1, but did not significantly affect GLUTs’ expression levels. Cana (100 µM) treatment reduced glycolysis, elevated the amino acids supplying the tricarboxylic-acid cycle, and significantly increased purine/pyrimidine-pathway metabolites, in contrast to dapa (3 µM) and ertu (10 µM). (4) The results confirmed cana´s inhibition of mitochondrial activity and GU at supra-pharmacological and pharmacological concentrations, whereas the dapa, ertu, and empa did not show effects even at supra-pharmacological concentrations. At supra-pharmacological concentrations, cana (but not dapa or ertu) affected multiple cellular pathways and inhibited GLUT1.

## 1. Introduction

The selective sodium-glucose co-transporter-2 inhibitors (SGLT2) canagliflozin (cana), dapagliflozin (dapa), ertugliflozin (ertu), and empagliflozin (empa) are approved as therapeutic glucose-lowering agents for the treatment of type 2 diabetes patients [1,2,3,4,5,6,7,8]. They show a 250-, 1200-, 2500-, and 2000-fold selectivity for SGLT2 over SGLT1 [9], and were developed based on the naturally occurring SGLT inhibitor, phlorizin. Phlorizin was isolated from apple-tree bark in 1835 [10], and since it was shown to normalize glycemia in 90% of pancreatectomized animals in the 1980s, interest in its potential to treat diabetes has grown [11]. However, phlorizin acts as a nonspecific SGLT inhibitor that is poorly absorbed from the gastrointestinal tract and not sufficiently stable for clinical use; therefore, it has not been considered for further clinical development [12].

Synthetic selective SGLT2 inhibitors, such as cana, empa, and others, have been developed and their clinical development has progressed. Cana, as a highly potent and selective SGLT2 inhibitor, has shown to lower the renal threshold for glucose, increase urinary glucose excretion, improve glycemic control and beta-cell function in rodent models of type 2 diabetes mellitus, and to reduce body-weight gain in rodent models of obesity [13]. Cana, the first SGLT2 inhibitor to be FDA-approved, in 2013 [14], and one of the most commonly used SGLT2 inhibitors, is about 5 to 10 times less selective for SGLT2 over SGLT1 than dapa, ertu, and empa [8]. Thus, cana may inhibit SGLT1, at approved therapeutic doses, in the small intestine [15].

Furthermore, sotagliflozin, a non-selective SGLT inhibitor acting on both SGLT1 and 2 [16], with strong affinity for SGLT2 and mild selectivity for SGLT1, has been developed and, recently, the results of two randomized controlled trials focusing on heart failure and renal outcomes of sotaglifozin-treatment were published [17,18].

SGLT1 is responsible for a major part of the dietary glucose uptake in the intestine [19], and the SGLT1-inhibitory effect gives the added value of delaying glucose absorption in the intestine and preventing post-meal hyperglycemia, [5,6,8,20,21,22,23]. However, although sotagliflozin has been approved in the EU since 2019 for overweight type 1 diabetic patients who do not achieve adequate glycemic control despite optimal insulin therapy [24], it is not yet approved by the FDA due to the increased risk of ketoacidosis [25].

SGLT2 is responsible for the majority of glucose reuptake in the tubular system of the kidneys [19], and selective SGLT2 inhibitors act by blocking the SGLT2-mediated reabsorption of glucose and sodium in the proximal tubule of the kidneys [26]. Apart from the glucose-lowering effect, SGLT2 inhibitors reduce body weight, blood pressure and, more importantly, the risk of cardiovascular events, heart-failure hospitalizations, and the progression of renal diseases [5,6,8,20,21,22,23]. Although the use of SGLT2 inhibitors is generally regarded as safe, cana has been associated with an increased risk of lower-limb amputations in type 2 diabetic patients [23,27,28,29,30]. Hence, discussions are ongoing as to whether the various SGLT2 inhibitors have different off-target effects at the cellular level.

To address potential differences in the clinical profiles between the SGLT2 inhibitors, several in vitro studies have investigated the cellular effects of SGLT2 inhibitors on energy metabolism [31,32,33,34], cellular function, proliferation and migration [35], and cell metabolism [33,36,37]. A common finding of these studies in different cell models is that cana exhibits an inhibitory effect on mitochondrial function [22,23,31,32,33]. Effects of dapa and empa on mitochondrial complex I activity have also been reported, although these are weak and cell-line-dependent [31,32,34].

Endothelial cells make up the lining of the vascular system, and their dysfunction may lead to complications that are observed in diabetes and other metabolic disorders [38,39,40,41]. The autonomous effects of SGLT2 inhibitors on endothelial cells have been investigated for their relevance to vascular function of SGLT2 inhibition [34,35,42]. The data from these studies showed that cana inhibited cell proliferation and glucose uptake (GU), while dapa and empa did not [35,42].

The inhibition of mitochondrial function by cana has also been observed in previous metabolomics studies on several cell lines, which showed alterations in the fatty-acid and purine/pyrimidine metabolism [33,36,37]. Dapa treatment has been shown to increase levels of amino-acid metabolites and to reduce levels of various lipid species [37].

However, currently, no data are available for the effects of different SGLT2 inhibitors on mitochondrial activity, cellular glucose uptake, and metabolic processes in endothelial cells; effects that could potentially influence the clinical profiles of these SGLT2 inhibitors.

Therefore, in this study, we aimed to investigate the effects of cana, dapa, ertu, and empa on mitochondrial oxidative phosphorylation and GU in human-umbilical-vein endothelial cells (HUVECs). To assess the putative impact of SGLT2 inhibitors on the function of the endothelium, we also examined the impact of the SGLT2 inhibitors cana, dapa, and ertu on the metabolomes of these cells.

## 2. Results

### 2.1. Effects on the Oxygen-Consumption Rate (OCR)

The HUVEC cells treated with a supra-pharmacological concentration (100 µM) of the SGLT2 inhibitor cana showed inhibition of OCR of nearly 60% (Figure 1A), similar to complex 1 inhibitor rotenone (2 µM). By contrast, dapa (100 µM) reduced the OCR by only 7%, while ertu and empa (both at 100 µM) had no significant effect on the OCR in the HUVECs compared to the DMSO (Figure 1A,B).

Next, we assessed the dose responses of the different SGLT2 inhibitors on the OCR. Cana inhibited the OCR in a concentration-dependent manner with a half-maximal inhibitory concentration (IC_50_) of 27 µM. The IC_50_ values for dapa, ertu, and empa were greater than 100 µM (data not shown). The OCR inhibition became apparent at concentrations above 10 µM of cana (Figure 1C), and at 100 µM, the effect on OCR was fully pronounced.

### 2.2. Effects on the Extracellular Acidification Rate (ECAR)

At supra-pharmacological concentrations (100 µM), cana, dapa, ertu, and empa lowered the ECAR by 45%, 13%, 32%, and 17%, respectively (Figure 2A,B). The IC_50_ values were 9 µM for cana and 39 µM for ertu. The IC_50_ values for dapa and empa were greater than 100 µM. At a concentration of 10 µM, the effect of cana on ECAR was less pronounced.

### 2.3. Effects on the ADP/ATP Ratio

The cana treatment of the HUVECs resulted in an increase in the ADP/ATP ratio at the concentration equal to the C_max_ elicited by its pharmacological dose (10 µM), and an even larger statistically significant response at the supra-pharmacological concentration (100 μM). The Dapa treatment, both at the pharmacologically relevant concentration of 0.3 µM and at the supra-pharmacological concentration of 3 μM, failed to cause any significant changes to the ADP/ATP ratio. However, the treatment with ertu at 1 µM (pharmacologically relevant concentration) and 10 μM displayed a weak, concentration-dependent increase in the ADP/ATP ratio (Figure 3). The rotenone increased the ADP/ATP ratio to 25.

### 2.4. Effects on the GU Using ^14^C-Deoxyglucose (^14^C-DOG)

Both cana and ertu inhibited ^14^C-DOG uptake with IC_50_ values of 14 µM and 85 µM, respectively (Figure 4A). At 100 µM, dapa and empa showed weak, but significant, increases in ^14^C-DOG uptake (dapa: *p* = 0.0411; empa *p* = 0.0003), with IC_50_ values > 100 µM (Figure 4A). The GLUT1 inhibitor, BAY-876 (25 µM), inhibited 60% of the ^14^C-DOG uptake and cana (100 µM) inhibited 97% of the ^14^C-DOG uptake (Figure 4B).

### 2.5. Effects of Cana on the GLUT Gene Expression

The treatment of the HUVECs with cana (10 and 100 μM) had no significant effects on the expression of the glucose transporters GLUT 1, 3, 6, 8, and 10, nor detectable cytotoxic effects (data not shown). The rotenone did not cause detectable cytotoxicity.

### 2.6. Metabolomics

The four main classes of ontology distribution for the MVA_UVA metabolites measured were amino acids, peptides and analogues (16 metabolites), nucleotides and analogues (21 metabolites), carbohydrates and conjugates (ten metabolites), and metabolites related to energy metabolism (seven metabolites). One cana (10 µM) sample was excluded due to a technical problem during the HPLC-MS measurement. Seventeen metabolites were altered significantly (*p*-value < 0.01) in the cana (100 µM)-treated HUVECs (Figure 5), while five metabolites for the ertu (10 µM)-treated and two metabolites for the dapa (3 µM)-treated cells showed at least low significance levels (*p*-value < 0.05). The same metabolites and metabolite classes were also changed in the HUVECs treated with a lower concentration of cana (10 µM), although fewer metabolites were affected (three metabolites, *p*-values ≤ 0.05). All the metabolite specifications are listed in detail in the Appendix A.

#### 2.6.1. Effects of Cana on the Levels of Amino Acid Supplying the Tricarboxylic Acid Cycle (TCA) Cycle

Cana at 100 µM significantly increased the levels of most of the amino acids supplying the TCA cycle (Figure 6), including glutamine (*p*-value: 0.036) and glutamate (*p*-value: 0.004) (Figure 7). Cana at 10 µM showed similar trends in changes in amino acid levels and increased aspartate levels. Ertu at 10 µM and dapa at 3 µM had no effects on the levels of these amino acids.

#### 2.6.2. Effects of Cana on Glycolysis Activity and Beta-Oxidation

The metabolites used for glycolysis were significantly depleted in the cana (100 µM)-treated HUVECs (*p*-value < 0.01; Figure 7).

The cana (100 µM)-treated HUVECs showed a trend towards reduced levels of free fatty acids (FFA). Cana at 10 µM showed the same trend, but less pronounced. The acetyl-CoA levels were significantly increased (*p*-value = 0.005) in the cana (100 µM)-treated HUVECs (Figure 5), whereas the treatment with cana at 10 µM induced a trend, albeit with no significance, towards increased acetyl-CoA levels. The supra-pharmacological concentrations of dapa and ertu (3 µM and 10 µM) had no discernible effect on the metabolites in the glycolysis.

#### 2.6.3. Effects on Redox Equivalents

At 100 µM, the cana significantly increased the NAD^+^ and decreased the NADH levels. The NAPDH levels were shifted in the direction of the NADP^+^. At 10 µM, cana depicted a trend towards increased NAD^+^ and decreased NADH levels, and the trend of the NADPH shifted to the NADP^+^ levels. The HUVECs treated with cana (100 µM) had a significantly increased ADP/ATP ratio and cana (10 µM) induced the same effect as the trend, but no significance was observed. Furthermore, no significant effect on redox equivalents was induced by the SGLT2 inhibitors ertu and dapa.

#### 2.6.4. Effects on Metabolites Used in Apoptotic Pathways

The levels of the apoptosis-associated metabolites, glutamine, glutamic acid, and n-acetyl-glutamic acid, were significantly increased in the cana (100 µM)-treated cells (*p* < 0.05). Cana (10 µM) showed a trend of increased glutamic acid and n-acetyl-glutamic acid, but no significance was perceived. No effects on apoptosis-associated metabolites were observed in the dapa- and ertu-treated HUVECs, irrespective of the applied SGLT2 inhibitor concentrations.

#### 2.6.5. Effects on Purine and Pyrimidine Pathway Activity

The cells treated with cana (both 10 and 100 µM) showed a trend towards increased levels of metabolites in the purine/pyrimidine pathways, whereas a trend towards reduced metabolite levels in the purine/pyrimidine pathways was seen for the cells treated with dapa (0.3 and 3 µM) and ertu (1 and 10 µM).

## 3. Discussion

In the present study, we investigated the differential cell autonomous effects of the selective SGLT2 inhibitors cana, dapa, ertu, and empa on parameters related to mitochondrial activity (OCR, ECAR, ADP/ATP ratio) and on GU in HUVECs. Further, a metabolomics analysis was performed on the cana-, dapa- and ertu-treated HUVECs to assess their effects on the metabolomes of these cells. Our results confirmed the effects of cana on the mitochondrial activity and GU in the endothelial cells, which have been described previously in other cell models, using supra-pharmacological and pharmacologically relevant concentrations. By contrast, dapa, ertu, and empa did not show any of these effects, even when applied at supra-pharmacological concentrations.

In more detail, we reproduced the effects of cana on the OCR in the HUVECs. At 100 µM, the cana inhibited the OCR by 60% in our study, and weak OCR inhibition was also observed after the treatment with cana at 10 µM, which was consistent with the reported results from several other cell lines [31,32,33]. In agreement with the results obtained in the renal proximal tubule epithelial cells [33], dapa and empa did not inhibit the OCR at concentrations of up to 100 μM in the HUVECs in our study.

The reduction in the OCR by cana was likely the result of the electron transport chain (ETC) complex 1 inhibition in our study. Nonetheless, when interpreting the results, one needs to take into account that endothelial cells, such as HUVECs, do not predominantly rely on their mitochondrial energy supply [44,45,46]. In fact, the majority of the energy supply in HUVECs is generated via anaerobic metabolism (e.g., glycolytic pathway), even in the presence of sufficient oxygen [43,45,47]. Thus, lactate production is sustained even during ETC complex 1 inhibition in HUVECs, and proangiogenic signaling is promoted [48]. Cana treatment (10 and 100 µM) seems to additionally diminish the already low-grade usage of oxygen for cellular ATP generation, and to shift the energy production even further in the direction of anaerobic metabolism, i.e., glycolysis. It is unclear whether these effects would translate to the in vivo situation, since they were only significant when applying 100 µM of the SGLT2 inhibitor, while only trends of these effects were observed in the cana (10 µM)-treated HUVECs.

ETC complex 1 inhibition by cana has also been reported in human renal proximal tubule epithelial cells [33]. Shetty et al. have observed that the inhibition of oxidative phosphorylation in liver cells resulted in a rapid stimulation of glucose transport mediated by the GLUT1 transporter, which occurred without a change in the GLUT1 or GLUT1 mRNA content of the cell [49]. In many cell types, GLUT1 is responsible for basal glucose uptake [50], with decreasing intracellular ATP concentrations leading to increased glucose transport via GLUT1 [51]. Rotenone has stimulated GU and glucose consumption in HepG2 and immortalized mouse myoblast cells when mitochondrial oxidative phosphorylation was inhibited [52]. This stimulatory effect led to increased lactate release and thus contributed to the increased acidification of the extracellular space [52]. We found that cana and ertu inhibited the GU in the HUVECs and that the GU decrease in the HUVECs by cana was caused by the inhibition of the GLUT1. The treatment of the HUVECs with cana did not have a significant effect on the expression of the GLUTs, but an effect of cana on the protein level cannot be excluded.

The metabolomics analyses of cana (10 and 100 µM), ertu (1 and 10 µM), and dapa (0.3 and 3 µM)-treated HUVECs showed that cana, in contrast to dapa and ertu, caused a significant decrease in the metabolites (*p* < 0.01) used in the glycolysis, which was in agreement with the lower GU observed after the cana treatments.

Low glucose levels cause enhanced mitochondrial respiration in endothelial cells, whereas high glucose levels induce reduced mitochondrial respiration, even when enough oxygen is present [42,44,53]. During the cana treatments (both 10 and 100 µM), the mitochondrial oxidative phosphorylation in the HUVECs was impaired, and the reduced glycolysis rate, together with the impaired mitochondrial oxidative phosphorylation, produced a metabolically quiescent state in the cells to conserve the cellular functions.

The decrease in the ECAR in the HUVECs induced by cana at 100 µM was consistent with the increased lactic acid levels observed in the metabolomics analyses. We assume that the lactate was retained in the cells for further use in the regeneration of NAD+ (anaerobic respiration via pyruvic acid).

The levels of the amino acids used for the production of glucose showed a trend towards elevated levels in the cana (10 µM)-treated HUVECs, and were significantly higher in the cana (100 µM)-treated cells, indicating reduced TCA-cycle activity due to its low usage in endothelial cells, as described previously [53], and/or that the metabolites and pathways supplying it were in a state of suspension due to the blockage of the ETC complex 1 and adjoining pathways.

A potential increase in the TCA-cycle activity could also have been caused by the increased activity of the electron transport chain complex II (ETC2), since ETC2 directly links the TCA cycle and the respiratory chain by catalyzing the oxidation of succinate to fumarate [54]. A potential increase in fatty acid oxidation could lead to heightened activity of the ETC2, as has been described by Pfleger et al. [55].

Secker et al. observed the inhibitory effects of cana (50 µM) on glutamate dehydrogenase (GDH) in human renal proximal tubule epithelial cells after treatment [33]. This effect was concurrent with the inhibition of ETC complex I and the replenishment of the TCA-cycle intermediates and intermediates of the amino acid metabolism. The increased levels of glutamine and glutamate also suggest GDH inhibition by cana in the HUVECs. In HUVECs, glutamate is used to produce gamma-aminobutyric acid via glutamic acid decarboxylase, while about one third of the TCA carbons are derived from glutamine [56].

The inhibition of glutamic acid decarboxylase reduces cellular ATP levels and increases pyruvate and fatty-acid oxidation. The blockage of the glutamine metabolism has also been shown to perturb proliferation dynamics and the adoption of a senescence-like phenotypes [57].

In agreement with the results from Hawley et al., who have studied human embryonic kidney cells [31], cana evoked a concentration-dependent increase in the ADP/ATP ratio in the HUVECs in our study. Since glucose is required for mitochondrial ATP production in endothelial cells [58], the inhibition of GU and mitochondrial oxidative phosphorylation would negatively affect the health status of endothelial cells in the long-term.

Trends in a cellular-depleted redox-state caused by lower levels of metabolites, such as NADPH and NADH, in the cana-treated HUVECs were observed. In particular, the NADPH, a crucial cofactor for endothelial nitric oxide synthase [59], was lowered in the cana-treated HUVECs, thus decreasing the functionality and affecting the cellular ability to deal with reactive oxygen species, which in turn would result in endothelial cell dysfunction.

The HUVECs treated with dapa (3 µM) and ertu (10 µM) showed no significant changes regarding glycolysis, the metabolites used in cell respiration and beta-oxidation. However, these cells showed reduced metabolite levels in the purine and pyrimidine pathways, and these effects were more pronounced with increasing SGLT2 inhibitor concentrations. The levels of metabolites from the purine and pyrimidine pathways were significantly lower after the ertu (10 µM)- and dapa (3 µM) treatments, which were similar to the results described in [37,60], than after the cana (10 µM)- and (100 µM)-treatments in the HUVECs. This reflects a significant difference in the mode of action of ertu and dapa compared to cana. A reduction in purine metabolism after the dapa treatment is described by Stack et al. [60], suggesting a reduction in cell death [60] and, potentially, an improvement in endothelial function [37]. The significant inhibitory effect and the observed differences in the mode of action of cana in relation to those of the other three investigated SGLT2 inhibitors in this study could have been caused by cana´s lower selectivity for SGLT2 over SGLT1 compared to dapa, ertu, and empa. This could be one of the reasons why cana inhibited the mitochondrial activity and GU at supra-pharmacological and pharmacological concentrations, whereas dapa, ertu, and empa did not show effects, even at supra-pharmacological concentrations.

## 4. Materials and Methods

### 4.1. SGLT2 Inhibitors and GLUT1 Inhibitor

The synthetic selective SGLT2 inhibitors, cana, empa, and ertu [9], were purchased from Ark Pharma (Arlington Heights, IL, USA). The synthetic selective SGLT2 inhibitor, dapa was obtained from AstraZeneca (Gaithersburg, MD, USA). Initial experiments were performed by applying supra-pharmacological concentrations of cana, dapa, ertu, and empa, and the subsequent experiments were performed with concentrations that equaled the plasma peak concentration (C_max_) of the respective SGLT2 inhibitor elicited by its pharmacological doses. The maximal plasma concentration (C_max_) of cana at daily therapeutic dose of 300 mg at steady state is approximately 10 µM (4445 ng/mL) [61,62], C_max_ of dapa at daily therapeutic dose of 10 mg at steady state is about 0.3 µM (122 ng/mL) [63], and 1 µM (436 ng/mL) is the C_max_ of ertu at daily therapeutic dose of 15 mg at steady state [64]. C_max_ of empa at steady state is 0.63 µM (284 ng/mL) at a daily dose of 25 mg [65].

In contrast to sodium-dependent glucose transporters (SGLT), glucose transporters (GLUT) are facilitative glucose transporters [66] that are overexpressed in many tumors [67]. BAY-876 is an orally active and highly selective glucose transporter 1 (GLUT1) inhibitor with an IC50 of 2 nM. BAY-876 is about 130-fold more selective for GLUT1 than GLUT2, GLUT3, and GLUT4, and a potent blocker of glycolytic metabolism and ovarian cancer growth [68].

### 4.2. Cell Lines and Cell Culture Conditions

HUVECs (Lonza, Switzerland) were plated and expanded in culture medium according to the manufacturer’s instructions (CC-5035 EGM-PLUS BulletKit Medium; CC-5036 EGM-PLUS Basal Media + CC-4542 EGM-PLUS SingleQuots Kit, Lonza, Switzerland).

### 4.3. Measurement of the OCR and the ECAR

Data were analyzed as change in OCR and ECAR as a function of SGLT2 inhibitor concentration. Effects on cellular respiration and medium acidification was measured with Agilent/Seahorse Extracellular Flux analyzer XF96 at 37 °C. Basal OCR and ECAR were recorded prior to injection of the respective SGLT2 inhibitor to yield final concentrations of 0.2, 1, 5, 25, and 100 µM (n = 4), followed by five repeated measurements of OCR and ECAR (during approximately 35 min). Rotenone (2 µM) was used as reference for the inhibition of the mitochondrial ETC complex I. DMSO was utilized as blank sample. In total, 15,000 and 25,000 HUVEC cells were seeded in 150-microliter cell culture medium per XF96 plate well 18–24 h prior to experiment. XF96 cartridge was equilibrated with calibration buffer overnight according to manufacturer´s instructions. Cell culture medium was changed to XF Base Medium Minimal DMEM (Agilent, Santa Clara, CA, USA) complemented with 5 mM glucose, 1 mM pyruvate, 1 mM glutamine, and 15 mM HEPES 30–40 min prior to first reading. After the last read-out, cells were fixed in 4% formaldehyde, and subsequently washed with 4% formaldehyde prior to adding Hoechst nuclear stain for cell counting with Image Xpress system from Molecular Devices. OCR and ECAR were normalized based on the cell numbers per well.

### 4.4. Measurement of the ADP/ATP Ratio

The cellular energy status was assessed by measuring the ADP/ATP ratio, following SGLT2 inhibitor treatment for three hours. ADP and ATP levels in the samples were detected using the ADP/ATP Ratio Assay Kit (ab65313, Abcam, Cambridge, UK). Cells were grown as described above. Prior to the measurement, the culture medium was removed from the plate, and Nucleotide Releasing Buffer (50 µL per 10^3^–10^4^ cells) was added. After five minutes of incubation at room temperature while gently shaking, a total of 100 µL of the reaction mix was added and background luminescence was assessed. Fifty µL of cells (10^3^–10^4^) treated with Nucleotide Releasing Buffer were pipetted into a luminometer plate and samples were read after approximately two minutes in a luminometer. A 10 × ADP converting enzyme was diluted 10-fold with Nucleotide Releasing Buffer. ADP levels in cells were read by adding 10 µL of 1 × ADP converting enzyme and reading the output after two minutes. All samples were run as duplicates in four independent experiments. ADP/ATP ratios were calculated based on the manufacturer´s recommendation.

### 4.5. Measurement of the GU

Effects on the GU were assessed using ^14^C-deoxyglucose (^14^C-DOG) as tracer. In total, 40,000 HUVECs per well were seeded in a Cytostar 96w SPA plate and allowed to adhere overnight prior to the experiment. Cytostar plates were pre-coated with collagen I (0.03 µg/uL). The cells were pre-treated for 15 min with 0.2, 1, 5, 25, and 100 µM of the respective SGLT2 inhibitor. Next, they were pre-diluted in assay medium consisting of XF Base MedSium Minimal DMEM (Agilent 102353) and complemented with 5 mM glucose, 1 mM Pyruvate, 1 mM Glutamine, and 15 mM HEPES prior to the addition of 18 µM ^14^C-DOG tracer (specific activity: 1 µCi/mL). Uptake of ^14^C-DOG was monitored in real-time using a micro beta scintillation counter (PerkinElmer) with four readings between 10 and 190 min. Cell monolayers were visually inspected for potential loss of adhesion after the last reading. A total of 25 µM BAY-876 (n = 16) was used as reference for effects on GLUT1-mediated uptake against cana (100 µM, n = 8) [68], and DMSO was used as control (n = 32). The ^14^C-DOG uptake rate was calculated as fmol DOG per minute, and after subtracting the value for ^14^C-background. Data were fitted to a four-parameter logistic model to calculate IC_50_.

### 4.6. RNA Isolation and cDNA Synthesis

Total RNA was extracted from HUVECs treated with cana (10 and 100 µM) after 4 and 24 h, and purified using the Qiagen RNeasy mini kit (Qiagen, Valencia, CA, USA). Cell lysis was performed in an RNases inactivating buffer provided by the manufacturer. Until RNA extraction, the samples were stored at −70 °C. RNA concentrations and 260/280 absorbance ratios were measured spectrophotometrically using Nanodrop ND-1000. Reverse transcription of the purified RNA was performed immediately after extraction. The cDNA was synthesized using a High-Capacity cDNA Reverse Transcription Kit (Applied Biosystems, 168 Third Avenue, Waltham, MA, USA). The reaction was set with 800 ng RNA in 20 µL of solution containing RT-random primers, 100 mM deoxyribonucleotide triphosphates (dNTPs) mix, 2.5 U/µL MultiScribe Reverse Transcriptase, and RT buffer. Real-time qPCRs were performed on Applied Biosystems^®^ QuantStudio™ at 50 °C for two minutes and at 95 °C for ten minutes. Then, denaturation cycles (40 cycles of 95 °C for 15 s) were performed. Subsequently, the annealing and extension were performed, during which fluorescence was measured (40 cycles of 60 °C for 60 s). Data expression levels were recorded as quantification cycles, CT. Data were acquired using real-time PCR software (QuantStudio™, Applied Biosystems, USA). The mean CT values of the triplicate reactions were used in the subsequent analysis. Protein Atlas was used for data comparison.

### 4.7. Metabolomics

Cultivated HUVECs (median cell number 3.12 × 10^5^) were treated with cana (10 µM and 100 µM), ertu (1 µM and 10 µM), and dapa (0.3 µM and 3 µM) and with vehicle solution (1 µM and 10 µM vehicle solution 1 × phosphate-buffered saline 1 × PBS) as controls with six replicates per treatment (n = 6).

Two hours after treatment, the cell culture samples were extracted, dipped into isotonic NaCl solution with 4.5 g/mL glucose and freeze-dried for 9.5 h, as described previously [69,70]. The cells were then reconstituted in 90 µL H_2_O. Twenty-five µL of each cell sample were used for the measurement and fifteen µL of each cell sample were used for the quality control sample pool (QC). The cell samples were divided into two smaller batches of 24 samples each, and these two batches were measured in one run. Maximum measurement time per batch was 24 h.

Samples, blanks (pure H_2_O), QCs, and UltimateMix (UM) were measured in a stratified randomized sequence and analyzed with a Dionex Ultimate 3000 HPLC (Thermo Fisher Scientific, USA) equipped with a reversed-phase Atlantis T3 C18 pre- and analytical column (Waters, USA) with an injection volume of 10 µL, as described previously [70,71,72]. Raw data were converted into mzXML (msConvert, ProteoWizard Toolkit v3.0.5) [73], and the detected m/z of known metabolites were identified using PeakScout [70,74,75] with a reference list containing respective mass and retention times.

Principal component analysis (PCA) was performed, centered and scaled to unit variance (R function prcomp). Missing values were imputed by regularized expectation-maximization (R function impute PCA, estim_ncpPCA). For ANOVA analysis, the treatment was taken as the fixed factor (simple ANOVA; R function gls, missing values were not imputed). For specific group comparisons, pairwise post hoc tests (R function lsmeans) were conducted. Technical variability was 15.9% median relative standard deviation (RSD) in the QC samples for 60 multivariate and univariate analysis (MVA_UVA) metabolites and yielded a further 36 univariate-analysis (UVA) metabolites. The MVA_UVA metabolites were classified according to the Human Metabolome Database [43].

### 4.8. Statistical Analyses

Results for OCR, ECAR, and GU were presented as mean ± SD. Results for ADP/ATP ratios were presented as mean ± SEM.

Data were analyzed for statistical significance using one-way analysis of variance (ANOVA) corrected for multiple comparisons by Tukey statistical hypothesis testing (GraphPad 8.0.1, San Diego, CA, USA). A *p*-value < 0.05 between groups was considered as statistically significant.

Statistical analysis for metabolomics was performed with R (R Core Team, v3.4.1, R Foundation for Statistical Computing, Vienna, Austria) and TibcoSpotfire (v7.5.0, TIBCO, Palo Alto, CA, USA). All metabolites were controlled for their analytical quality [70] and graded into two classes: (1) suitable for multivariate analysis and univariate analysis (MVA_UVA) and (2) suitable for univariate analysis (UVA). Quality assessment was performed according to [70], and a separate control for outliers due to sequence position, time-point of measurement, replicate number, cell count, date of sampling, measurement batch, and sample extraction events was conducted via PCA.

To correct for cell number differences and reduce technical variability, median normalization was performed. Each metabolite was normalized to the sample median using the following procedure. First, each metabolite´s peak area was scaled between 0 and 1 within one sample. Next, the median of all 0-to-1-scaled metabolites’ peak areas within one sample was calculated, and each peak area was divided by the sample median value.

Finally, normalized peak areas were log-transformed. Kolmogorov–Smirnov and Brown–Forsythe Levene tests were used to discern normality and homoscedasticity of data distribution. The log10-transformed, normalized data were normally distributed according to the Kolmogorov–Smirnov test (93.8% of all metabolites were normally distributed) and homoscedastic according to the Brown–Forsythe-Levene-type test (92.7% of all metabolites were homoscedastic). The *p*-values corrected for multi-parameter testing via Benjamini–Hochberg are given in the Appendix A. All values and additional information are reported in Appendix A. 

## 5. Conclusions

Our data confirmed the reported inhibition of mitochondrial activity by cana at supra-pharmacological and, to some extent, at pharmacological concentrations. By contrast, dapa, ertu, and empa did not show any measurable effect on the mitochondrial activity, glucose uptake, and ECAR in the endothelial cells, even at concentrations that were much higher than the C_max_ evoked by their pharmacological doses. The supra-pharmacological concentrations of cana, but not dapa, ertu, or empa, affected multiple cellular pathways (glycolysis, beta-oxidation, and cellular energy state) and blocked the GLUT1.

At 100 µM, cana showed a combined reduction in glycolysis and mitochondria respiration. This diminished cellular-energy state could lead to insufficient angiogenesis [52] and could play a role in microvascular complication, through involvement in, for example, impaired wound healing in people with type 2 diabetes.

## Figures and Tables

**Figure 1 ijms-23-07966-f001:**
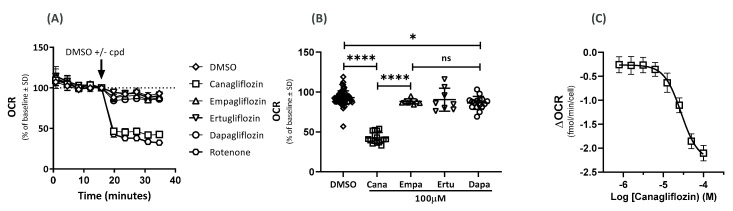
Effects of SGLT2 inhibitors on OCR. Effect of DMSO; DMSO + SGLT2 inhibitor (100 µM); cana, dapa, ertu, and empa on OCR and rotenone (2 µM) in (**A**) OCR as % average change from baseline and (**B**) OCR after 40 min (n = 16 (cana and dapa), n = 8 (ertu and empa), and n = 32 (DMSO); **** *p* < 0.0001, * *p* = 0.041, ns = not significant, n = replicates of samples). (**C**) ΔOCR for cana (n = 12).

**Figure 2 ijms-23-07966-f002:**
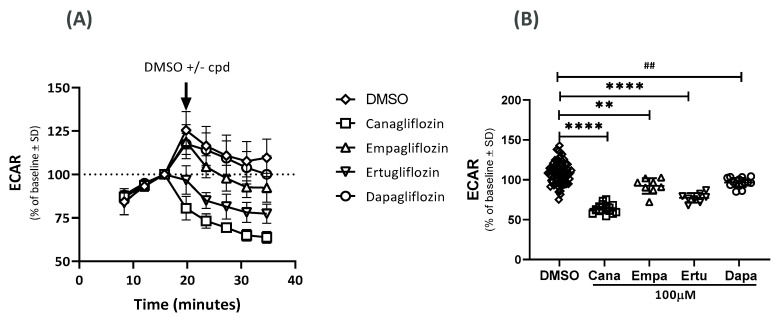
Effects of SGLT2 inhibitors on ECAR. Effect of DMSO and DMSO + SGLT2 inhibitor (100 µM); cana, dapa, ertu and empa on (**A**) ECAR as % change from baseline (one experiment). (**B**) ECAR after 40 min (two independent experiments); **** *p* < 0.0001; ** *p* = 0.0021; ## *p* = 0.0026; n = 62 (DMSO), n = 16 (cana), 15 (dapa) and 8 (ertu and empa).

**Figure 3 ijms-23-07966-f003:**
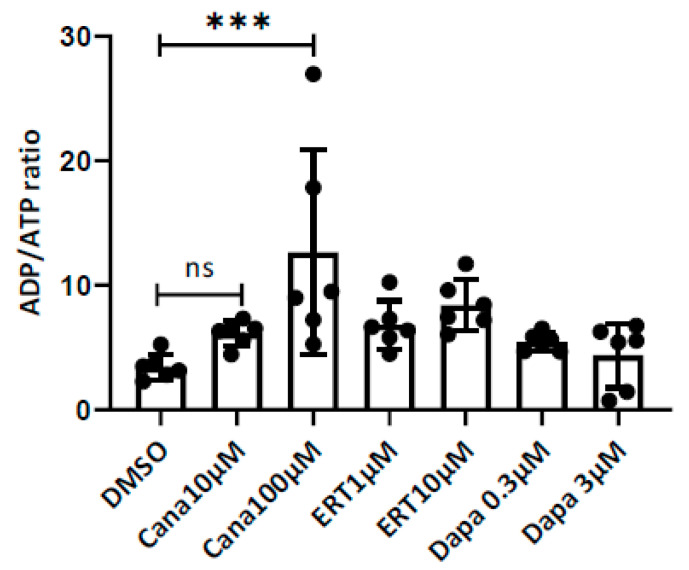
Effects of SGLT2 inhibitors on the ADP/ATP ratio. Effect of DMSO and DMSO + SGLT2 inhibitor on the ADP/ATP ratio in HUVECs at pharmacologically relevant and supra-pharmacological concentrations of cana (10 and 100 µM), dapa (0.3 and 3 µM), ertu (1 and 10 µM). *** *p* < 0.004, ns = not significant. Ordinary one-way analysis of variance (ANOVA).

**Figure 4 ijms-23-07966-f004:**
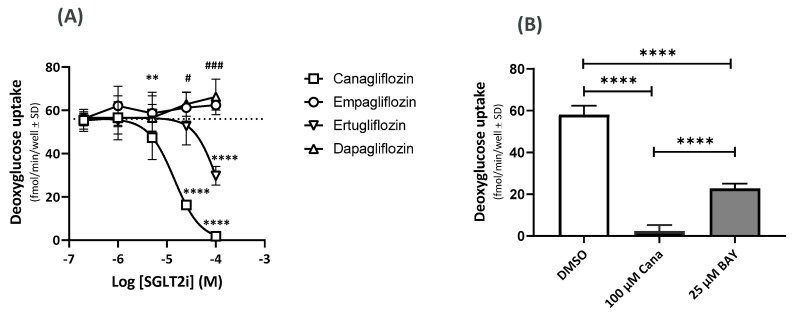
Effect of SGLT2 inhibitors on 14C-DOG uptake (**A**) and inhibition of GU by cana and BAY-876 (**B**). (**A**) 14C-DOG uptake for cana, dapa, ertu, and empa and in controls (dotted lines). ** *p* = 0.0082 (cana vs. DMSO), **** *p* < 0.0001 (cana and ertu vs. DMSO), # *p* = 0.0411 (dapa vs. DMSO), ### *p* = 0.0003 (dapa vs. DMSO); n = 28 (DMSO), n = 11–12 (cana), n = 11–12 (dapa), n = 7–8 (ertu), and n = 4 (empa). (**B**) Inhibitory effects on GU in the presence of cana (100 µM) and GLUT1 inhibitor, BAY-876 (25 µM). **** *p* < 0.0001 (n = 32 (DMSO), n = 8 (cana), and n = 16 (BAY-876)). Effects of cana on GLUT gene expression.

**Figure 5 ijms-23-07966-f005:**
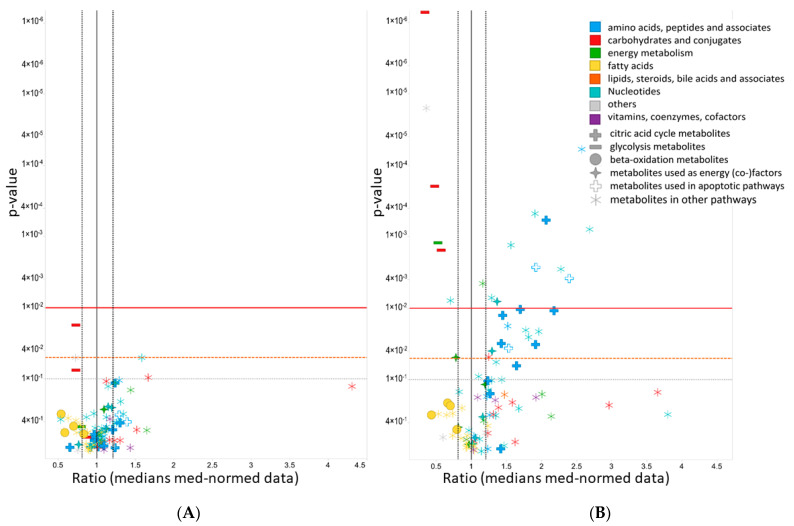
ANOVA volcano plots for cana (10 µM)-treated (**A**) and cana (100 µM)-treated HUVECs (**B**). ANOVA exhibited significant differences between the control groups for 96 metabolites. Levels of metabolites on the left side of the black vertical line decreased; levels of metabolites on the right side of the black vertical line increased. *Y*-axis: *p*-values (inverse log scale); *x*-axis: metabolite ratios (median normalized data; simple treatment). Left black dotted vertical line: ratio of 0.8. Right black dotted vertical line: ratio of 1.2. The red horizontal line denotes a *p*-value of 0.01, the dashed orange horizontal line denotes a *p*-value of 0.05, and the gray horizontal line denotes a *p*-value of 0.1. Metabolites are colored according to the Human Metabolome Database classes [43]. Metabolites supplying the TCA cycle are marked with filled crosses; glycolysis metabolites by filled minus; metabolites used in beta-oxidation with filled circles; metabolites used as energy (co-)factors with filled stars; metabolites used in apoptotic pathways are marked with empty symbols and others with snowflake-like symbols.

**Figure 6 ijms-23-07966-f006:**
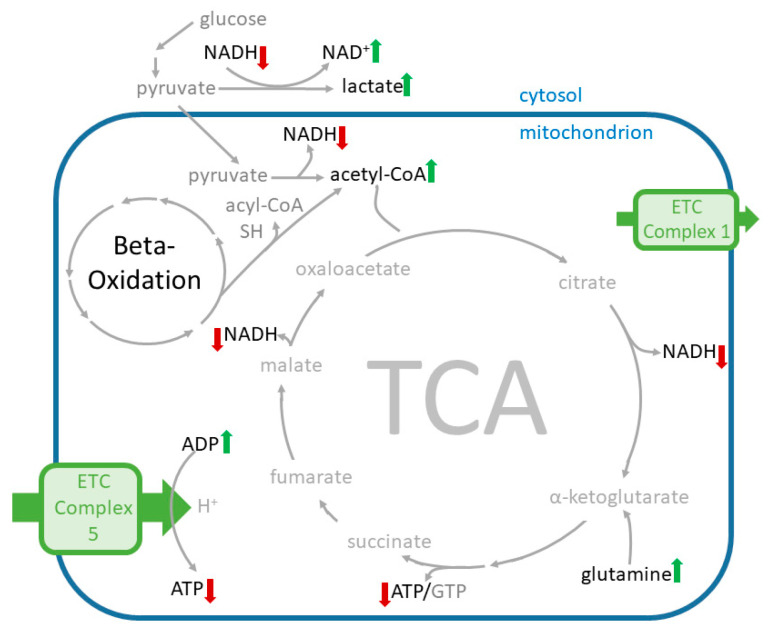
Scheme of the affected metabolites and pathways in cana-treated HUVECs. Metabolite levels significantly elevated by cana (100 µM) and changed in the same direction by cana (10 µM) are marked with a green up-arrow, metabolite levels lowered by cana (100 µM) are marked with a red down-arrow.

**Figure 7 ijms-23-07966-f007:**
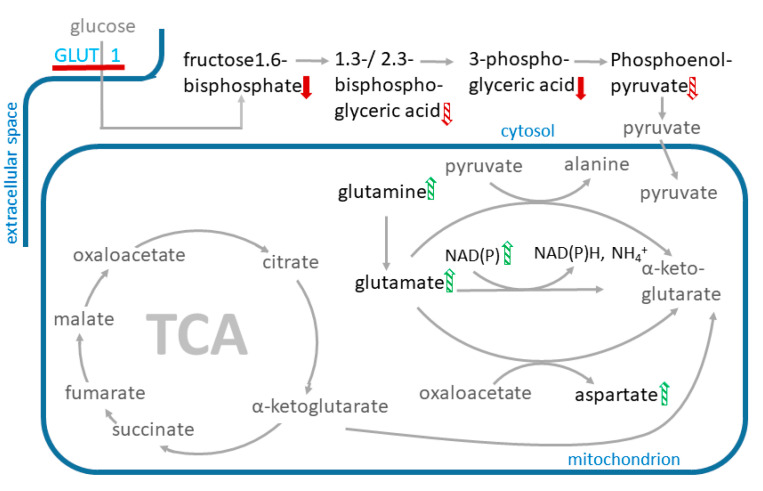
Scheme of affected metabolites and pathways in cana-treated HUVECs. Metabolite levels that were elevated after cana treatment are marked with a green up-arrow, metabolite levels lowered after cana treatment are marked with a red down-arrow. Metabolites that appear more than once in the graph are only marked once with an arrow. Significantly changed metabolite levels in cana (10 µM)- and cana (100 µM)-treated cells are marked with a filled arrow and metabolite levels that are significantly changed only in cana (100 µM)-treated cells are marked with a shaded arrow. The red line at the GLUT 1 transporter indicates transporter inhibition.

## Data Availability

The data presented in this study are available on request from the corresponding authors.

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
