# Peer review of "Differential In Vitro Effects of SGLT2 Inhibitors on Mitochondrial Oxidative Phosphorylation, Glucose Uptake and Cell Metabolism"

_ijms, 2022, doi:10.3390/ijms23147966_

Round 1
Reviewer 1 Report
This is a straightforward comparison of several of the biochemical effects of canagliflozin, dapagliflozin, ertugliflozin, and empagliflozin on on mitochondrial oxidative phosphorylation, glucose uptake and cell metabolic products in human umbilical endothelial cells. At suprapharmacologic concentration and to a limited extent pharmacologic concentration, canagliflozin exerted effects while the other SGLT2 inhibitors did not. That finding is not altogether surprising because canagliflozin also possesses significant SGLT1 inhibitory effect, which could be playing a role. It is somewhat surprising that the authors did not include phlorizin in their study, but we cannot ask them to now repeat all their experiments. Nonetheless, they should amply discuss known phlorizin data in their introduction and background and in their discussion. It would have been interesting to include sotagliflozin as one comparator, but, again, we cannot ask them to repeat all their experiments. They should at least discuss sotagliflozin as far as they are able.
SPECIFIC COMMENTS
1) As far as supra versus pharmacologic concentrations, they must add results driving from known phramacokinetics of the various SGLT2 inhibitors. In other words, typical plasma concentrations of these agents.
2) Please give a better description of BAY-876 which is a GLUT-1 inhibitor used in several of their experiments.
Reviewer 2 Report
The study touches on a very important topic of the features of the action and possible mechanisms of the formation of side effects when using SGLT2 inhibitors. The work is certainly very relevant and sheds light on the important features of the formation of side effects of canagliflozin.
However, a more thorough preparation of the article for publication is required, as there are comments and suggestions:
Pages 4-5. - Figure 2 is repeated twice in different versions and with different signatures.
Page 7 – The figure needs to be enlarged, since the data of the lower third are very poorly distinguishable.
Page 8 – lines 165-166 – correction of references to figures is required. Figures 3 and 4 follow Figure 5.
Figure 4 is located unsuccessfully – the reference to it is in section 2.6.1, and the figure is in section 2.6.2.
It may be worth noting that the role of complex II is not discussed at all, although it is known that the suppression of complex I leads to the intensification of complex II. Moreover, the authors found significantly increased levels of most amino acids that provide the TCA cycle.
Reviewer 3 Report
In their manuscript X. Peng, C. Magnes et al. report on the in-cellular effects of SGLT2 inhibitors on mitochondrial functionalities. In general, the experimental strategy is designed adequately and the conclusions drawn from enough number of different experiments are fully consistent. The work could gain some more points if some experiments on mitochondrial membrane potential were conducted, however, the authors mentioned that such experiments were reported in the literature.
The negative part: the manuscript looks like a draft of very low quality. For instance, all figures have a very bad resolution of bold(?) micromolar sign. Moreover, something wrong with Figure 2, which is presented in duplicate. The first appearance of Figure 2 is with poor description. Right after Figure 2 the authors followed up with Figure 5, then Figure 3, then Figure 4. There are two “p-value” axis names in figure 5. This is (maybe?) acceptable for journals with IF 0.2-0.3, but not with IF of 6.
The conclusions from statistical experiments are somewhat I would call anti-scientific. All values in the work are given without standard deviation, and comparisons are provided with “p values” only, without a number of repetitions. The worst part is in phrases like “Only arginine showed slightly elevated levels in ertu (10 μM)-treated HUVECs (p-value 0.121).”, line 169 (similar context is on lines 151 and 193). If the p-value is higher than 0.05, you cannot drop the Null hypothesis. As a result, it is scientifically incorrect to make any conclusions about the difference between these 2 groups.
Based on a very bad manuscript design and scientifically wrong statistical judgments I could imagine the quality of experimental work. This is a reason why I do not think that this work is worth to be published.
Round 2
Reviewer 1 Report
The paper is now significantly improved. Would add plasma concentrations in
ng/ml as well as molar. Here is a citation to support the issue of pharmacological concentration:
Iijima H, Kifuji T, Maruyama N, Inagaki N. Pharmacokinetics, Pharmacodynamics, and Safety of Canagliflozin in Japanese Patients with Type 2 Diabetes Mellitus. Adv Ther. 2015 Aug;32(8):768-82. doi: 10.1007/s12325-015-0234-0. Epub 2015 Aug 18. PMID: 26280756; PMCID: PMC4569680.
Reviewer 3 Report
Based on the quality of the original manuscript, I doubt that the removal of unpleasant statements will automatically improve the quality of the experimental work done. I still suggest resubmitting the paper to a journal with a less significant IF.
